# Football Match Line-Up Prediction Based on Physiological Variables: A Machine Learning Approach [†]

Alberto Cortez [1,*], António Trigo [1,2] and Nuno Loureiro [3,4]

1. Coimbra Business School Research Centre | ISCAC, Polytechnic of Coimbra, 3045-601 Coimbra, Portugal; antonio.trigo@gmail.com
2. ALGORITMI Research Center, University of Minho, Campus de Azurém, 4800-058 Guimarães, Portugal
3. Sport Sciences School of Rio Maior, 2040-413 Rio Maior, Portugal; nunoloureiro@esdrm.ipsantarem.pt
4. Life Quality Research Centre (CIEQV), Polytechnic Institute of Santarem, 2040-413 Rio Maior, Portugal
* Correspondence: alberto.v.cortez@protonmail.com
† This paper is an extended version of our paper published in Cortez, A.; Trigo, A.; Loureiro, N. Predicting Physiological Variables of Players that Make a Winning Football Team: A Machine Learning Approach. In Proceedings of The 21st International Conference on Computational Science and Its Applications (ICCSA) 2021, Cagliari, Sardinia, Italy, 13–16 September 2021; pp. 3–15.

**Abstract:** One of the great challenges for football coaches is to choose the football line-up that gives more guarantees of success. Even though there are several dimensions to analyse the problem, such as the opposing team characteristics. The objective of this study is to identify, based on the players' physiological variables collected using Global Positioning Systems (GPS), which players are the most suitable to be part of the starting team/line-up. The work was developed in two stages, first with the choice of the most important variables using the Recursive Feature Elimination algorithm, and then using logistic regression on these chosen variables. The logistic regression resulted in an index, called the line-up preparedness index, for the following player positions: Fullbacks, Central Midfielders and Wingers. For the other players' positions, the model results were not satisfactory.

**Keywords:** football; line-up; physiological variables; predicting; machine learning; classification; logistic regression

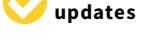



## 1. Introduction

It is broadly known that technology fosters not only knowledge but also a deep understanding of science with major impact on society. Sports is no different as it benefits from the use of technology [1]. Intelligent systems open new possibilities regarding professional sports [2], with data-driven methods being able to effectively overcome limitations related to the individual match analysis subjectivity while offering better comprehension of the game for football teams [3]. Football is a good example of a sport that could benefit from the integration of Artificial Intelligence (AI) in its analysis [4].

GPS devices have become a commonplace in professional football to track the player's performance. The use of such devices allows the identification of internal and external load measures that can provide answers about which variables to include in an integrated approach [5]. As the amount of information collected by these GPS is significantly increasing, its analysis becomes increasingly complex, becoming crucial to create analytical algorithms that can effectively analyse the data.

Data science has emerged as a strategic area that, supported by the great volume of data retrieved from within the competition environment (training and games), allows for knowledge discovery in sport science with the aim of filling some gaps that traditional statistical methods could not achieve [6]. The enormous advances in technology make it possible to process a great amount of data while enabling the drawing of extremely useful

conclusions [2]. One such area where predictive systems have gained a lot of popularity is in the prediction of football match results [2].

With the purpose of identifying the best line-up for a football match a study was performed using Machine Learning (ML) algorithms on a dataset composed of the physiological variable values of football players recorded using a GPS system. The data used in this work was taken during football training sessions and matches of a Portuguese team from the 2nd Regional Division of AF Santarém in the 2018/2019 season, using Playertek (Catapult Innovations, Melbourne, Australia [7]) with a sampling frequency of 10 HZ.

The work presented in this paper is an extension of the paper "Predicting physiological variables of players that make a winning football team: a machine learning approach" presented by the authors at ICCSA 2021 conference. The objective of the work presented at the conference was to identify if there was a strong correlation between the football match outcome (wins) and the physiological variables of the players, having achieved an accuracy of 79% with the XgB algorithm [8].

The present paper is organised as follows. The next section summarises the relevant literature on ML in football. Then the methodology is described, to which follows the presentation and discussion of the main findings and conclusions. Finally, the main contributions, limitations, and suggestions for future research are addressed.

## 2. Related Work

Artificial Intelligence (AI) presence in sports is gradually increasing, in particular in football, where ML algorithms are used to detect meaningful patterns based on positional data [6]. ML is already used in football to predict and prevent injuries in players [5,9,10], as well as in the categorization of football players and football training sessions [1,11], in the evaluation of football players regarding their market value [12,13], and in predicting the results of football matches [4,14], among others.

Regarding injury risk, several studies aimed to understand whether it is possible to improve the ability of a neuromuscular screen to identify injury risk factors in elite male youth football players using ML algorithms [5], while other studies aimed to compare different ML methods to choose the injury risk factor model to identify athlete at risk for lower extremity muscle injuries, for this it was used a dataset with 132 male professional players from football and handball during the season 2013/2014 [9]. In this work, the best performing algorithm to identify players at risk was Alternating Decision Tree with an area under curve, 0.747; true positive rate, 65.9; true negative rate, 79.1. In [10] authors used the data collected from a GPS, subjective questionnaires, and the injury data from 40 elite male players over one season, to predict non-contact injuries. To predict 1-week injuries internal load features were more accurate and to predict 1-month injuries, a combination of internal and external loads achieved the best performance.

In terms of categorization, there are also some important studies. In [11] authors aimed at identifying technical-tactical behaviours of players using their statistics, without including spatial-temporal descriptors by using ML methods. In this work, authors assessed the capability of ML to identify the most influential variables for each of the positions on the field and to find groups of outlier players. Another model regarding categorization is proposed in [1] aiming at evaluating football training sessions using AI, through the analysis of the evaluation indexes of the running ability of athletes at different positions. This work shows that the evaluation efficiency is 24.12% higher than that of a traditional artificial team, proving the feasibility of this model.

In [12], authors propose a novel method for estimating the value of players in the transfer market, based on the FIFA 20 dataset. The dataset was clustered using the APSO-clustering algorithm which resulted in detecting four clusters: goalkeepers, strikers, defenders, and midfielders. Then, for each cluster, an automatic regression method, able to detect the relevant features, was trained, and they were able to estimate the value of players with 74% accuracy. In this line, authors from [13] presented a study where ML techniques are explored to plan possible player transfers and build a professional football team with

interesting results (accuracy = 0.82, precision = 0.84, recall = 0.82, and F1-score = 0.83). A dataset with the complete overview of a football player was used: technical and physical parameters, but also his psychological state in order to improve the scouting of players and to understand if it was possible to create the definition of a successful transfer (the results were not constant for all experiments).

In terms of predicting game outcomes there are different approaches. Either using a dataset from English Premier League (EPL) with featured engineering and exploratory data analysis to select a group of variables to create an improved set (with the most important features) [4], although the results obtain could not outperform the bookmarkers prediction. In [14], authors used other datasets besides EPL, namely, training data from Japan, New Zealand, Mexico, South Africa, Russia, and other countries. To evaluate the model, RPS (Rank Probability Score) function was selected to determine the predictive accuracy of the models. and to the EPL much lower results than to the lowest leagues analysed. Still regarding the prediction of the match results, Stübinger et al. [15] applied ML techniques in football betting using players characteristics, the dataset was composed by all the matches from the top 5 leagues and the corresponding second leagues between 2006 and 2018. In predicting the game win, Random Forest was the best performing algorithm with 81.26%, and achieved an economical significant return of 1.58%.

As illustrated by these examples, ML algorithms hold enormous potential to provide coaching staff in a football team with additional information to evaluate the game [16].

## 3. Methodology

The aim of the present work is to develop an information system, using artificial intelligence, to identify potentially football match line-ups, which architecture is presented in Figure 1. The system collects the physiological variables from the GPS vests of the players. These variables are imported into a database, to which some transformations are performed. The ML model is used for calculating the line-up preparedness index of each player, which then the soccer coach can consult and use in his decision making.

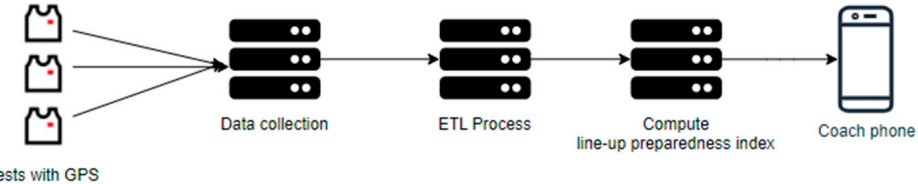

**Figure 1.** Proposed architecture.

For the ML process the Cross Industry Standard Process for Data Mining (CRISP-DM) [17] process was selected, as in [18] but in sport result prediction. The CRISP-DM framework has six phases: business understanding, data understanding, data preparation, modelling, evaluation, and deployment. Apart from the deploy stage, which was not performed since the model has not yet been put into production, all the others were performed and are presented in the following subsections.

### 3.1. Business Understanding

The purpose of this work is to construct a player fitness index to help football coaches choose the players who should be starters, i.e., those who should participate in the starting line-up.

To create this index, all the physiological variables of each player were collected by GPS both in games and in training. In addition to this data, the results of the team in the games (target variable) and the initial line-up of the players in the games were also recorded.

To build the index several steps were followed: first, the physiological variables of the players, by field position (Central Defender (CD), Full Back (FB), Centre Midfielder (CM), Offensive Midfielder (OM), Winger (W) and Forward (F)), that most contribute to victory were identified using Recursive Feature Elimination (RFE) [19,20] (already carried

out in a previous study [8]. Then ML algorithms were trained to determine if the variables selected by the RFE algorithm produced better results than using all variables. Finally, a logistic regression was performed on the most appropriate set of variables to create the fitness index of the players to be part of the team taking the field.

As a result, it is expected that this predictive index can have an influence on the training analysis, because the coaching staff can understand where they should apply more focus in the training sessions to perform better on game day.

### 3.2. Data Understanding

For this study, a dataset was compiled with the physiological variables of football players recorded by a GPS during the training sessions and football games of a Portuguese team from the 2nd Regional Division of AF Santarém in the 2018/2019 season (see Table 1).

**Table 1.** Variables in each dataset.

| Games Dataset | Training Sessions Dataset |
|---|---|
| *'Athlete', 'Game', 'Position', 'Home or Away', 'Pitch', 'Final Score', 'Minutes', 'Game Condition', 'RPE_J', 'sRPE_J', 'HR', '%HR', '<60%HR', '60–74.9%HR', '75–89.9%HR', '>90%HR', 'Player Load', 'Player Load.UA/min', 'Distance_m', 'Distance.m/min', 'Distance.0–3', 'Distance.3.4', 'Distance.4–5.5', 'Distance.5.5–7', 'Distance.>7', 'WRRatio', 'Accel.0–2', 'Accel.2–4', 'Accel.>4', 'Deacc.0–2', 'Deacc.2–4', 'Deacc.>4'* | *'Athlete', 'Game', 'Position', 'Home or Away', 'Pitch', 'Final Score', 'Minutes', 'RPE_J', 'Player Load', 'Player Load.UA/min', 'Distance Total', 'Distance.m/min', 'Distance.0–3', 'Distance.3.4', 'Distance.4–5.5', 'Distance.5.5–7', 'Distance.>7', 'WRRatio', 'Accel.0–2', 'Accel.2–4', 'Accel.>4', 'Deacc.0–2', 'Deacc.2–4', 'Deacc.>4'* |

In addition to the physiological variables, the dataset regarding the football matches also contains the line-up selection for the games, contextual variables such as the type of field where the training session/match was played (Pitch) and definition of home or away game (Home or Away) and the games result variable (Final Score), which will be, after its transformation, our target variable (win), that will hold the indication whether the game was won or not (binary).

During the 2018/2019 the team with 28 players with an average age of 22 years old played 14 games in the first phase of the championship, 10 games in the second phase and 2 extra games in Ribatejo Cup, regarding the training session, 39 training sessions were recorded through 13 microcycles. A total of 33,748 different episodes were registered, regarding the different players, and games they played. In addition, the dataset regarding the training sessions had a total of 24,360 different episodes, regarding the different players. ML algorithms were applied to these records, which results are presented in Sections 3.4.1 and 3.4.2.

Table 2 presents the description of the variables used in this study.

### 3.3. Data Preparation

At this stage the selected data was processed to be later used by the ML models. Firstly, null or inconsistent information was removed from the dataset. After cleaning the data two transformations were performed. The first was to create a *Win* variable (data labelling) since the objective of this study are the victories and therefore the distinction between draw and defeat was not considered relevant in the scope of this work. In this case, the variable *Win* contains the information about the victory in the game (1—victory; 0—other). This variable was computed from the variable with the game result, "Final Score" (see Tables 1 and 2). The second one was to merge the different datasets, associating the variables *win* and *line-up* of the games with the training sessions, as this dataset does not contain these two variables. In order to do so, two new variables were created in the training sessions dataset namely, *win* and *line-up*. These new variables are used for the identification within the training dataset which players were in the starting 11 of each game and its outcome. To create this new variable in the training sessions dataset a query (join) that related the two

datasets using the variables *athlete* and *game* was built resulting in a new dataset with 16,996 episodes (see Tables 3 and 4). During the model and evaluation phases, the episodes were grouped by player position to run the different ML algorithms. The episodes regarding the Goalkeeper position were excluded because of the specific physiological demands regarding this position.

**Table 2.** Description of the variables in the datasets.

| Variables | Description |
|---|---|
| Game | Is the number of the game in the Championship games sequence. |
| Final Score | Represents the points that the team won in the game, which relates to a victory, draw or loss. 0 it's for a lost game, 1 for a draw and 3 for a win. |
| Minutes | It is the number of minutes that the players are actively in the game. |
| RPE | It is the rate of perceived exertion, made by a numeric estimate of someone's exercise intensity. It is a way to measure how hard a person is exercising, which ranges from 1 (no exertion) to 10 (extremely hard). |
| Heart Rate (HR) | The maximum heart rate calculated as HRmax = 220—age. It is calculated in absolute and %. |
| Player Load | Calculated based on the acceleration data that are registered by the triaxial accelerometers. This variable, considered as a magnitude vector, represents the sum of the accelerations recorded in the anteroposterior, medio-lateral and vertical planes. Represented in Total and in Arbitrary Units (U.A.) per minutes. |
| Distance (Total and m/s) | The Total distance provides a good global representation of volume of exercise (walking, running) and is also a simple way to assess individual's contribution relative to a team effort. It is divided in five different speed zones: "walking/jogging distance, 0.0 to 3.0 m/s; running speed distance, 3.0 to 4.0 m/s; high-speed running distance, 4.0 to 5.5 m/s; very high-speed running distance, 5.5 to 7.0 m/s; and sprint distance, a speed greater than 7.0 m/s [21]. |
| Work Ratio (WRRatio) | It is used to describe footballer's activity profiles, which is divided in two categories: (1) pause if the distance is travelled at a speed <3.0 m/s; and as (2) work if the distance is travelled at a speed >3.0 m/s. |
| Acceleration (Accel.) | Categorized based upon the acceleration of the movement, which is thought to represent the "intensity" of the action. It is divided in "low intensity", 0.0 to 2.0 m/s$^2$; "moderate intensity", 2.0 to 4.0 m/s$^2$; and "high intensity", greater than 4.0 m/s$^2$ [21]. |
| Deacceleration (Deacc.) | Categorized based upon the deacceleration of the movement, which is thought to represent the "intensity" of the action. It is divided in "low intensity", 0.0 to −2.0 m/s$^2$; "moderate intensity", −2.0 to −4.0 m/s$^2$; and "high intensity", greater than −4.0 m/s$^2$ [21]. |

Source: [8].

**Table 3.** Variables in Training Sessions Dataset with Line-up Identification.

| Training Sessions Dataset with Win and Lineup Identification |
|---|
| *'Athlete', 'Week', 'Positionx, 'Home or Away', 'Pitch', 'Final-Score', 'Minutes', 'Player_Load', 'Player Load_UA/min', 'Distance_m', 'Distance.m/min', 'Distance_0_3', 'Distance_3_4', 'Distance_4_5.5', 'Distance_5.5_7', 'Distance_>7', 'WRRatio', 'Aceler_0_2', 'Aceler_2_4', 'Aceler_>4', 'Desac_0_2', 'Desac_2_4', 'Desac_>4', 'Win', 'Line-up'* |

### 3.4. Modelling

The modelling stage was divided into two phases: one where the most important physiological variables of the players in determining the victory in the game are identified, and a second in where an index of the players best suited for the game from the point-of-view of physiological variables is generated—the so-called the line-up preparedness index—using logistic regression.

**Table 4.** First rows of training sessions dataset with Win and Line-up variables.

|  | Athlete | Code | Position | Home or Away | Pitch | Final-Score | Minutes | Player_Load | Player Load _UA/min | Distance_m | Distance.m/min | Distance_0_3 |
|---|---|---|---|---|---|---|---|---|---|---|---|---|
| 1 | 9 | 13 | 7 | 1 | 0 | 3 | 96 | 363.8 | 3.789583 | 5728.3 | 59.669792 | 3959.3 |
| 2 | 9 | 13 | 2 | 1 | 0 | 3 | 76 | 302.0 | 3.973684 | 5025.7 | 66.127632 | 3362.4 |
| 3 | 9 | 13 | 7 | 1 | 0 | 3 | 86 | 336.9 | 3.917442 | 4554.2 | 52.955814 | 3981.6 |
| 5 | 2 | 13 | 3 | 1 | 0 | 3 | 96 | 230.9 | 2.397917 | 4800.7 | 50.007292 | 3917.0 |
| 6 | 2 | 13 | 3 | 1 | 0 | 3 | 76 | 261.9 | 3.446053 | 5300.5 | 69.743421 | 4163.1 |

| Distance_3_4 | Distance_4_5.5 | Distance_5.5_7 | Distance_>7 | WRRatio | Aceler_0_2 | Aceler_2_4 | Aceler_>4 | Desac_0_2 | Desac_2_4 | Desac_>4 | Win | Line-up |
|---|---|---|---|---|---|---|---|---|---|---|---|---|
| 684.1 | 502.8 | 447.7 | 134.4 | 9.1 | 101 | 95 | 39 | 89 | 110 | 30 | 1 | 1.0 |
| 665.1 | 607.7 | 320.8 | 69.7 | 11.1 | 69 | 109 | 26 | 108 | 93 | 16 | 1 | 1.0 |
| 335.9 | 221.6 | 15.1 | 0.0 | 7.3 | 188 | 113 | 24 | 143 | 142 | 21 | 1 | 1.0 |
| 502.5 | 287.6 | 72.5 | 21.0 | 8.5 | 78 | 81 | 10 | 104 | 63 | 9 | 1 | 1.0 |
| 667.7 | 417.7 | 52.0 | 0.0 | 12.8 | 122 | 83 | 6 | 121 | 83 | 13 | 1 | 1.0 |

3.4.1. Selection of the Best Set of Variables

At this stage, two models were used to select the variables: one with all the variables available in the game dataset (see Table 1) and another where the RFE algorithm was used to determine the most important variables for determining the victory, for each player position. The variables selected by RFE allow the understanding of the different effects of a football game in the different positions, and how these variables affect winning the game. These variables are dependent on the player's position, displaying differences depending on the position as they are related to specific demands of the player function and team strategy. The results are presented in Table 5 and were obtained in a previous study conducted by the authors [8].

**Table 5.** Selected variables by player position after running the RFE algorithm.

| Position | Features/Variables |
|---|---|
| Central Defender (CD) | *'Distance.m', 'Distance.0–3', 'Distance.3–4', 'Distance.>7'* |
| Full Back (FB) | *'<60.0%HR', '60–74.9%HR', 'Player Load.UA/min', 'WRRatio'* |
| Central Midfielder (CM) | *'Player Load.UA/min', 'Distance.m/min', 'WRRatio', 'Aceler.>4'* |
| Offensive Midfielder (OM) | *<'60.0%HR', '60–74.9%HR', 'Player Load.UA/min', 'WRRatio'* |
| Winger (W) | *'<60.0%HR', '60–74.9%HR', '75–89.9%HR', 'Player Load.UA/min'* |
| Forward (F) | *'Distance.m', 'Distance.4–5.5', 'Distance.>7'* |

Source: [8].

The use of the model presented in Table 5 in the training session dataset faced/encountered two difficulties associated with missing values: the inexistence of data for the OM, which forced it to be removed from the model; and the inexistence of values regarding the variables associated with Heart Rate (*HR, %HR, <60%HR, 60–74.9%HR, 75–89,9%HR and >90%HR*), which mainly affected the position W that was left with only one variable to perform the logistic regression, so the literature was used to select a second variable and it was selected a variable defended by Altavilla [22] where it affirms that total distance during the game (*'Distance_m/min'*) is a very important variable for analysing the performance. The model resulting from these changes is shown in Table 6.

**Table 6.** Selected variables by player position without Heart Rate variables.

| Positions | Features/Variables |
|---|---|
| CD | *'Distance_m', 'Distance_0_3', 'Distance_3_4', 'Distance_>7'* |
| FB | *'Player Load_UA/min', 'WRRatio'* |
| CM | *'Player Load_UA/min', 'Distance_m/min', 'WRRatio','Aceler_>4'* |
| W | *'Player Load_UA/min', 'Distance_m/min'* |
| F | *'Distance_m', 'Distance_4_5.5', 'Distance_>7'* |

Then the two models, with and without RFE selected variables, were trained using ML algorithms (Decision Tree Classifier (DT) and Naïve Bayes Classifier (NB)), which results are presented in Table 7. The classification algorithms were selected with a purpose of analysing the possibility of predicting the win, because our target variable (win), is a binary variable (win or not win).

**Table 7.** Results of model 1 (all variables) and model 2 (using RFE).

| ML Algorithms and Models/Position | | CD | FB | CM | W | F |
|---|---|---|---|---|---|---|
| DT Accuracy with CV | Model 1 (all variables) | **83** | 69 | **83** | 63 | **70** |
| | Model 2 (using RFE) | **83** | **71** | 70 | **73** | **70** |
| NB Accuracy with CV | Model 1 (all variables) | 59 | **77** | 59 | 68 | 46 |
| | Model 2 (using RFE) | **73** | 72 | **65** | 68 | 57 |

To analyse the results, the previous table was divided for the two classification algorithms and each model. The present results of the ML algorithms for the 2 models were achieved with Cross-Validation (CV). 5-fold Cross Validation was used.

With the DT classification algorithm, the first model had better results for the CM (achieved 83% accuracy) and the same result for CD (with 83% accuracy) and F (with 70% accuracy). Regarding the other two positions, FB and W, performed better with the second model, achieving 71% and 73% for each position, in terms of accuracy.

Regarding the NB classification algorithm, the first model only achieved better results to one position, FB, with 77% accuracy. It achieved the same results for both models for the W position, with 68% accuracy. In addition, performed better with the second model, for the remaining positions, CD (73%), CM (65%) and F (57%).

Although the differences were not very significant, it was possible to confirm from the results the advantage of using the model with the most important variables for each position using the RFE algorithm compared to the model with all variables (see Table 7). The use of fewer variables makes it easier for the football coach to monitor them, focusing on the physiological variables that are most relevant to each player's position on the field. Thus, the variables from the second model were used in the logistic regression, the second part of the study.

### 3.4.2. Predicting the Starting Line-Up and Chose the Better Prepare Players

To predict the starting line-up a logistic regression was initially used. A logistic regression is a useful way to create a model of probability of a certain class or event to exist, resulting in an index. To create this, it was important to understand if the logistic regression could be applied to all the positions regarding the model created. So, the logistic regression was only applied to the positions where the variables achieved a $p$ value < 0.05. For that reason, only the results for FB (Table 8), CM (Table 9) and W (Table 10) are presented.

**Table 8.** Logistic Regression for Full Backs (FB).

|  | coef | Std err | z | $P > |z|$ | [0.025 | 0.975] |
|---|---|---|---|---|---|---|
| const | 2.0983 | 1.577 | 1.331 | 0.183 | −0.992 | 5.189 |
| *Player Load_UA/min* | −1.0209 | 0.582 | −1.753 | 0.080 | −2.162 | 0.121 |
| *WRRatio* | 0.1600 | 0.094 | 1.702 | 0.089 | −0.024 | 0.344 |

**Table 9.** Logistic Regression for Central Midfielders (CM).

|  | coef | Std err | z | $P > |z|$ | [0.025 | 0.975] |
|---|---|---|---|---|---|---|
| const | −3.0404 | 1.799 | −1.690 | 0.091 | −6.567 | 0.486 |
| *Player Load_UA/min* | 2.3050 | 0.784 | 2.939 | 0.003 | 0.768 | 3.842 |
| *Distance.m/min* | −0.0994 | 0.064 | −1.550 | 0.121 | −0.225 | 0.026 |
| *WRRatio* | 0.1218 | 0.121 | 1.006 | 0.314 | −0.116 | 0.359 |
| *Aceler_>4* | 0.0198 | 0.024 | 0.813 | 0.416 | −0.028 | 0.068 |

**Table 10.** Logistic Regression for Wingers (W).

|  | coef | Std err | z | $P > |z|$ | [0.025 | 0.975] |
|---|---|---|---|---|---|---|
| const | −3.2661 | 1.978 | −1.651 | 0.099 | −7.143 | 0.611 |
| *Player Load_UA/min* | 3.1864 | 1.074 | 2.967 | 0.003 | 1.082 | 5.291 |
| *Distance.m/min* | −0.0938 | 0.046 | −2.040 | 0.041 | −0.184 | −0.004 |

### 3.5. Evaluation

As for the players' initial line-up forecast for the match, as mentioned in the modelling section, a logistic regression was used initially to construct an index to create a model of probability.

Based on logistic regression, an index was created to help select the players for the matches, based on the values of the physical variables of the week of training prior to the

match, and understand if the player chosen for a particular match was the best prepared one. Using this index, the tables with the index values per player per microcycle (which includes the training sessions of the week) were created for the positions of the players where the regression was considered valid: FB (Table 11), W (Table 12) and CM (Table 13).

**Table 11.** Full Backs (FB) Index Analysis.

| | Victory | Index | ID/LUp/Week | | Victory | Index | ID/LUp/Week |
|---|---|---|---|---|---|---|---|
| 44 | 1 | 0.784477 | (8, 0, 2) | 68 | 0 | 0.776008 | (19, 0, 8) |
| 59 | 1 | 0.865374 | (8, 0, 2) | 45 | 1 | 0.511412 | (19, 0, 13) |
| 79 | 1 | 0.827929 | (8, 0, 6) | 33 | 1 | 0.612893 | (19, 0, 13) |
| 72 | 1 | 0.526791 | (8, 0, 6) | 83 | 0 | 0.406908 | (19, 1, 9) |
| 67 | 1 | 0.662985 | (8, 0, 6) | 10 | 0 | 0.560884 | (19, 1, 9) |
| 21 | 1 | 0.911756 | (8, 1, 3) | 31 | 0 | 0.442858 | (19, 1, 9) |
| 95 | 1 | 0.911697 | (8, 1, 3) | 58 | 1 | 0.440744 | (19, 1, 10) |
| 56 | 1 | 0.847468 | (8, 1, 3) | 53 | 1 | 0.493869 | (19, 1, 10) |
| 30 | 1 | 0.726883 | (8, 1, 4) | 88 | 1 | 0.608787 | (20, 0, 7) |
| 34 | 1 | 0.875619 | (8, 1, 4) | 71 | 1 | 0.568071 | (20, 0, 7) |
| 52 | 1 | 0.917653 | (8, 1, 7) | 6 | 1 | 0.877902 | (20, 0, 7) |
| 0 | 1 | 0.667928 | (8, 1, 7) | 77 | 1 | 0.675493 | (20, 0, 11) |
| 75 | 1 | 0.672547 | (8, 1, 7) | 74 | 0 | 0.748045 | (20, 1, 1) |
| 4 | 0 | 0.837648 | (8, 1, 8) | 2 | 0 | 0.621270 | (20, 1, 1) |
| 36 | 0 | 0.783271 | (8, 1, 9) | 43 | 0 | 0.313663 | (20, 1, 1) |
| 51 | 1 | 0.790515 | (8, 1, 11) | 57 | 1 | 0.673096 | (20, 1, 2) |
| 60 | 1 | 0.947780 | (8, 1, 12) | 35 | 1 | 0.602226 | (20, 1, 2) |
| 3 | 1 | 0.857310 | (8, 1, 12) | 25 | 1 | 0.595407 | (20, 1, 2) |
| 23 | 1 | 0.792565 | (8, 1, 12) | 16 | 1 | 0.695457 | (20, 1, 3) |
| 18 | 1 | 0.847074 | (8, 1, 13) | 66 | 1 | 0.654608 | (20, 1, 4) |
| 55 | 0 | 0.735844 | (10, 0, 8) | 13 | 1 | 0.557984 | (20, 1, 5) |
| 48 | 0 | 0.653187 | (10, 0, 8) | 86 | 0 | 0.656770 | (20, 1, 8) |
| 20 | 0 | 0.833750 | (10, 1, 1) | 11 | 0 | 0.779103 | (22, 0, 1) |
| 82 | 0 | 0.866102 | (10, 1, 1) | 9 | 0 | 0.282544 | (22, 0, 1) |

The tables presented were divided into four columns and all are important for understanding the data and the proposed analysis. The first column displays the index in the training session dataset. The second column shows whether the match outcome in which the player participated was a win or not. The third column is the value of the logistic regression index. Finally, the fourth column displays three values: "ID" that represents the identity of the player; "LUp" that indicates if this player was in the line-up; and "Week" that represents the number of the microcycle. To better understand the information associated with each table, two examples of correct decisions (circled in green) and one incorrect decision (red circled) will be identified for each table.

The table for the FB position is shown in Table 11, where correct decisions are highlighted in green and the wrong ones in red.

For example, the first correct decision according to the model is related to player ID 8 (in the records 21, 95 and 56) since in the three training sessions of the third week, the player's index reached 91% in the first two and 84% in the last training session. These results indicate that the player in question should be selected for the line-up, which in fact happened, resulting in a positive outcome for the match (win). In the same group, looking at player ID 20 (in the records 88, 71 and 6), it is possible to see that the player in question obtained two training sessions with a low index value (60% and 56%) and a good last training session (with 87%). According to the coaching staff, the player was not selected for the starting line-up, which according to the model is correct. The third player analysed in this example was the same player, player with the ID 20 (in the records 57, 35

and 25) regarding the second week. In this case, the player was selected for the starting line-up by the coaching staff, presenting an index of 67% in the first training session, 60% in the second training session and 59% in the third session. According to the proposed model, this decision did not reflect an adequate evaluation of the readiness/fitness of the player to be in the starting line-up. Table 12 presents the table regarding the Winger position.

**Table 12.** Wingers (W) Index Analysis.

| | Victory | Index | ID/LUp/Week | | Victory | Index | ID/LUp/Week |
|---|---|---|---|---|---|---|---|
| 44 | 1 | 0.985863 | (1, 0, 13) | 48 | 1 | 0.783514 | (11, 1, 7) |
| 32 | 0 | 0.600585 | (4, 0, 1) | 61 | 1 | 0.738240 | (11, 1, 7) |
| 20 | 0 | 0.480366 | (4, 0, 1) | 8 | 0 | 0.744943 | (11, 1, 8) |
| 60 | 0 | 0.515228 | (4, 0, 8) | 7 | 0 | 0.643265 | (11, 1, 8) |
| 11 | 0 | 0.663983 | (4, 0, 9) | 26 | 0 | 0.596132 | (11, 1, 8) |
| 3 | 1 | 0.657554 | (6, 0, 10) | 30 | 0 | 0.732079 | (11, 1, 9) |
| 53 | 1 | 0.708375 | (6, 0, 11) | 51 | 0 | 0.663642 | (11, 1, 9) |
| 22 | 1 | 0.937456 | (9, 1, 5) | 50 | 1 | 0.678929 | (11, 1, 10) |
| 5 | 1 | 0.863028 | (9, 1, 5) | 59 | 1 | 0.690924 | (11, 1, 10) |
| 24 | 1 | 0.784041 | (9, 1, 6) | 4 | 1 | 0.790619 | (11, 1, 11) |
| 21 | 1 | 0.933873 | (9, 1, 6) | 34 | 1 | 0.656216 | (11, 1, 11) |
| 54 | 1 | 0.873258 | (9, 1, 6) | 9 | 1 | 0.284175 | (11, 1, 11) |
| 39 | 0 | 0.904690 | (9, 1, 8) | 18 | 1 | 0.754153 | (11, 1, 12) |
| 31 | 1 | 0.848315 | (9, 1, 11) | 40 | 1 | 0.654352 | (11, 1, 13) |
| 45 | 1 | 0.472807 | (9, 1, 11) | 41 | 1 | 0.941015 | (11, 1, 13) |
| 0 | 1 | 0.955910 | (9, 1, 12) | 15 | 1 | 0.893009 | (20, 0, 6) |
| 36 | 1 | 0.929533 | (9, 1, 13) | 33 | 1 | 0.806206 | (20, 1, 5) |
| 57 | 1 | 0.952612 | (9, 1, 13) | 29 | 0 | 0.886793 | (22, 0, 9) |
| 38 | 1 | 0.679002 | (11, 0, 5) | 10 | 0 | 0.849515 | (22, 0, 9) |
| 28 | 1 | 0.668134 | (11, 0, 5) | 37 | 1 | 0.897781 | (22, 1, 3) |
| 12 | 0 | 0.527733 | (11, 1, 1) | 6 | 1 | 0.891282 | (22, 1, 3) |
| 2 | 0 | 0.668352 | (11, 1, 1) | 14 | 1 | 0.942967 | (22, 1, 3) |
| 56 | 1 | 0.677689 | (11, 1, 2) | 46 | 1 | 0.898490 | (22, 1, 4) |
| 13 | 1 | 0.840461 | (11, 1, 3) | 25 | 1 | 0.890783 | (22, 1, 4) |
| 35 | 1 | 0.724676 | (11, 1, 3) | 42 | 1 | 0.862396 | (22, 1, 10) |
| 43 | 1 | 0.782688 | (11, 1, 4) | 19 | 1 | 0.905162 | (22, 1, 10) |
| 55 | 1 | 0.783562 | (11, 1, 4) | 58 | 1 | 0.893174 | (22, 1, 10) |
| 16 | 1 | 0.815601 | (11, 1, 6) | 17 | 0 | 0.574255 | (27, 1, 1) |
| 49 | 1 | 0.555276 | (11, 1, 6) | 23 | 1 | 0.494040 | (27, 1, 2) |
| 47 | 1 | 0.855227 | (11, 1, 6) | 52 | 1 | 0.766669 | (27, 1, 2) |

For the Winger position, let us start by analysing the player with the ID 11 (in the records 38 and 28) in the fifth week of training. In this training week, the player only took part of the first and last session of the week with a reported index of 67% and 66%, respectively. In this case, the coaching staff did not select the player for the starting line-up, which according to the model was a correct decision. Another choice of the coaching staff aligned with the model proposed, is related to the player with the ID number 22 (in the records 37, 6 and 14). In this example, the player displayed good index values in week three, achieving 89%, 89% and 94%. In this case, the player in question was selected for the starting line-up, with the team having a positive outcome for the same week match (win). Another example chosen was the player with ID number 27 (in the records 23 and 52), for the second week. In this case, the player was selected for the starting line-up, but only achieved 49% in the first training session and 76% in the last one. These are considered low index values and for that reason and regarding the model is considered a poor choice, as it did not optimise the overall performance of the team.

**Table 13.** Central Midfielders (CM) Index Analysis.

| | Victory | Index | ID/LUp/Week | | Victory | Index | ID/LUp/Week |
|---|---|---|---|---|---|---|---|
| 44 | 1 | 0.784477 | (8, 0, 2) | 49 | 1 | 0.695287 | (14, 1, 7) |
| 59 | 1 | 0.865374 | (8, 0, 2) | 17 | 0 | 0.432993 | (14, 1, 8) |
| 79 | 1 | 0.807929 | (8, 0, 6) | 7 | 0 | 0.577226 | (14, 1, 8) |
| 72 | 1 | 0.526791 | (8, 0, 6) | 64 | 0 | 0.786492 | (14, 1, 9) |
| 67 | 1 | 0.662985 | (8, 0, 6) | 96 | 0 | 0.500672 | (14, 1, 9) |
| 21 | 1 | 0.911756 | (8, 1, 3) | 85 | 0 | 0.583720 | (19, 0, 8) |
| 95 | 1 | 0.911697 | (8, 1, 3) | 68 | 0 | 0.776008 | (19, 0, 8) |
| 56 | 1 | 0.847468 | (8, 1, 3) | 45 | 1 | 0.511412 | (19, 0, 13) |
| 30 | 1 | 0.726883 | (8, 1, 4) | 32 | 1 | 0.612893 | (19, 0, 13) |
| 34 | 1 | 0.875619 | (8, 1, 4) | 83 | 0 | 0.406908 | (19, 1, 9) |
| 52 | 1 | 0.917653 | (8, 1, 7) | 10 | 0 | 0.560884 | (19, 1, 9) |
| 0 | 1 | 0.667928 | (8, 1, 7) | 31 | 0 | 0.442858 | (19, 1, 9) |
| 75 | 1 | 0.672547 | (8, 1, 7) | 58 | 1 | 0.440744 | (19, 1, 10) |
| 4 | 0 | 0.837648 | (8, 1, 8) | 53 | 1 | 0.493869 | (19, 1, 10) |
| 36 | 0 | 0.783271 | (8, 1, 9) | 88 | 1 | 0.608787 | (20, 0, 7) |
| 51 | 1 | 0.790515 | (8, 1, 11) | 71 | 1 | 0.568071 | (20, 0, 7) |
| 60 | 1 | 0.947780 | (8, 1, 12) | 6 | 1 | 0.877902 | (20, 0, 7) |
| 3 | 1 | 0.857074 | (8, 1, 12) | 77 | 1 | 0.675493 | (20, 0, 11) |
| 23 | 1 | 0.792565 | (8, 1, 12) | 74 | 0 | 0.748045 | (20, 1, 1) |
| 18 | 1 | 0.847074 | (8, 1, 13) | 2 | 0 | 0.621270 | (20, 1, 1) |
| 55 | 0 | 0.735844 | (10, 0, 8) | 43 | 0 | 0.313663 | (20, 1, 1) |
| 48 | 0 | 0.653187 | (10, 0, 8) | 57 | 1 | 0.673096 | (20, 1, 2) |
| 20 | 0 | 0.833750 | (10, 1, 1) | 35 | 1 | 0.602226 | (20, 1, 2) |
| 82 | 0 | 0.866102 | (10, 1, 1) | 25 | 1 | 0.595407 | (20, 1, 2) |
| 80 | 0 | 0.411594 | (10, 1, 1) | 16 | 1 | 0.695457 | (20, 1, 3) |
| 19 | 1 | 0.866759 | (10, 1, 3) | 66 | 1 | 0.654608 | (20, 1, 4) |
| 38 | 1 | 0.865714 | (10, 1, 3) | 13 | 1 | 0.557984 | (20, 1, 5) |
| 93 | 1 | 0.865563 | (10, 1, 4) | 86 | 0 | 0.656770 | (20, 1, 8) |
| 61 | 1 | 0.909490 | (10, 1, 4) | 11 | 0 | 0.779103 | (22, 0, 1) |
| 29 | 1 | 0.947054 | (10, 1, 5) | 9 | 0 | 0.282544 | (22, 0, 1) |
| 5 | 1 | 0.787528 | (10, 1, 5) | 69 | 0 | 0.865051 | (22, 0, 1) |

Finally, the table regarding the CM is presented in Table 13.

Finally, in the case of Central Midfielders, the first example is related with player with the ID number 10 (in the records 93 and 61). The player performance was analysed in the two training sessions regarding the fourth week, where this player achieved 86% in the first training session and 90% in the last training session. In this training week, the player was selected for the starting line-up, with the team having won the match. Considering the model proposed, the option made by the coaching staff was good, since the index values are high (above 85%). The second player analysed in this example was the player with the ID number 19 (in the records 45 and 32). Considering the player performance index (51% and 61%) during the training sessions of week 13, the player should not be selected for the starting line-up, which is in agreement with the decision made by the coaching staff. The last example presented in this work refers to the player with the ID number 20 (in the records 57, 35 and 25) whose performance was analysed for the second week. In this example the player was selected for the line-up, and achieved 67% in the first training, 60% in the second training session, and 59% in the last training session. Considering the proposed model, this player did not display a good performance (low index values), even though the team won the week match.

## 4. Discussion

One of the first objectives was to understand if the physiological variables of the players (through their position on the field), using ML algorithms, could predict the game outcome. This first part of the work consisted in comprehending the possibility of predicting the outcome of a football match based on the physiological variables of the players. In a second part of this study, an RFE algorithm was used to choose the most important variables for each position on the field, and ML algorithms were run and compared with the results acquired for all the variables and the RFE variables. The results obtained with the model produced by RFE algorithm (see Table 6) were superior to the results obtained with the model with all variables (see Table 7), so this second model was considered for the second objective of the study.

It is critical to understand the variables selected and analyse them in the context of previous research. The variables selected in this work to be studied were identified from previous studies in football, these are considered the most important variables for each position on the field. As presented in this study, distance covered at max speed (*distance.>7*) by F was one of the variables that had a bigger impact on winning. As suggested in [23,24], teams with F's that covered more distance at a very high speed were closer to winning the match when comparing two opposing teams. For the Central Midfielders (CM) position, one of the variables selected was the distance covered per minutes as suggested in [22,25], and which refer that CM, during the matches, has the highest value in total distance covered. Regarding the Central Back (CB) position, one of the most important variables is the covered the shorter distances during match [25], and for that reason one of the variables selected for the CB position was Distance 0.3. The variable Player Load U.A./min was also considered an important variable in this study as it was selected by the RFE for the FB and W positions, not being selected for the CB. This was corroborated in a study performed by Baptista et al., where they argue that CB had less turns per match than the FB and W [24]. Another important conclusion from these studies is that FB covered more high intensity and sprinting distance than CB during matches [24]. This was also corroborated in this study because Work-Ratio was selected as one of variables for the FB.

The second part of this study was focused on creating a line-up preparedness index to help coaching staff decide which player should be in the starting line-up, or which player is better prepared to help the team win the game, concerning the player's physiological variables. Once again, the lack of sufficient data meant that the regression model built was only valid for three positions (FB, W and CM). However, for those three positions interesting results were obtained. These results open an interesting opportunity to understand the choices made by the coaching/technical team. Allowing the technical staff to evaluate their choices and performance of the team and if these decisions are in accordance with the index created (which represents the intensity of the value of the physiological variables selected for the different positions of the players), in other words if the best prepared players by the coaching staff were chosen.

Despite the numerous challenges of obtaining complete datasets and improved statistics, the results presented here already provide an interesting look at the players selection process. Nevertheless, it is easy to understand that the results of football games do not depend exclusively on the players' physiological variables, but also on technical and tactical issues of both the players and the opposing teams, which were not addressed in the context of the present work.

## 5. Conclusions

Current football competition has the characteristics of fierce confrontation, long duration, high intensity, and high specialised and strategic requirements. On these attributes, it is easy to understand that football is a profoundly complex game. Choosing and assessing properly the team players performance through their physiological and physical variables is essential. This is also an important and difficult task in a multiplayer sport such as football, that has a multitude of aspects that need to be considered [2]. Therefore, studies in

this area might assume a significant role in terms of evaluating the potential performance of teams and players in a competing scheme, where individual performance plays an important role. Additional vital information can be retrieved from the individual data, for instance, where it can be used for predicting players performance and injury probability.

The results obtained in this study can be used by the coaching staff of a football team in their preparation. Additionally assisting with recognizing by player's position on the field and understanding which are the main factors for the players' performance. Bottom line, the utilisation of ML applied to training and selection can address some of the current challenges of the sport. The use of ML in such a context represents a significant step forward in the game knowledge, providing possibly critical information to the training staff. The possibility of making available this critical information will permit the assistance of professionals in the decision-making process, improving simultaneously their understanding of the competitive physiological demands of the different field positions and game moments/situations.

This study was performed in just one football team, and for that reason limited in terms of the information that can be retrieved. Despite being an important limitation to the study, it can serve as a stepping stone to different investigations that may help to foster the knowledge of the game. In the future, we plan to further analyse different teams in different contexts and therefore enhance our sample set. This will allow a deeper understanding and to acquire a more complete image of what the most important variables are and how these can be related to the strategic choices to the teams.

As defended in [26], the future of team performance will be, more than ever before, based on data insights. Nonetheless, it is critical to comprehend that when a human component is associated with sport, there will always be unpredictability, making its outcome in general intriguing and surprising for its followers [27].

**Author Contributions:** Conceptualization, A.C., A.T. and N.L.; formal analysis, A.C. and A.T.; investigation, A.C., A.T. and N.L.; methodology, A.C., A.T.; resources, N.L.; validation, A.C. and A.T.; writing—original draft: A.C. and A.T.; writing—review and edit: A.C., A.T. and N.L. All authors have read and agreed to the published version of the manuscript.

**Funding:** This research received no external funding.

**Institutional Review Board Statement:** Not applicable.

**Informed Consent Statement:** Not applicable.

**Data Availability Statement:** Data is contained within the article.

**Conflicts of Interest:** The authors declare no conflict of interest.

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
