# Peer review of "Football Match Line-Up Prediction Based on Physiological Variables: A Machine Learning Approachâ€"

_computers, doi:10.3390/computers11030040_

Round 1
Reviewer 1 Report
The authors propose a system to predict the lineup of a football team on the basis of a variety of observed variables (most of them by means of GPS sensors). A subset of useful variables is selected by means of the well known Recursive Features Elimination methodology, then logistic regression is used to learn a fitness index for a player. Experiments have been conducted and good results are obtained for some player positions.
This work is interesting and well described. I only have a minor point the authors should discuss. Logistic regression works with linear combination of variables, why the authors do not consider other more complex machine learning models such as neural networks, random forests, etc. ?
Reviewer 2 Report
This paper shows an interesting application using machine learning. However, there are some issues that should be addressed as shown below.
Introduction
- The introduction was so small, it is recommended to give the motivation for this interesting application.
Related work
- It is better to move the first paragraph of related work to the introduction since I gave a good motivation for this application. Line 45 to 51.
- It is more useful to mention the authors, their works, and the reported results for the paragraph in lines 62 to 65. Like what the author mentioned in the next paragraph.
- In categorization, there are some studies. Such as? Line 68.
Methodology
- In line 104. Is there any related work that shows how much this method works properly?
Data Preparation
- It is recommended to call the process in the paragraph starts in line 139 as data labeling, not a transformation.
Modeling (It was written as Modelling!)
- In line 188. The method name was wrong, it is Naive Bayes classifier, not Naïve Baines Classifier.
- For table 6. There are two methods of ML where be used with and without feature selection. Does the author try to fuse the scores? In this regard, the author will achieve two goals. The first goal with end up with one final score, the second one, the author will improve the result.
Discussion
- In line 313. The word “and” should be detected!
Reviewer 3 Report
The manuscript requires a lot of improvement in its current state. Even though the work's novelty is subpar, the study is interesting and has practical applications. My corrections are as follows:
- Please add a flowchart of the approach - end to end
- Merge tables 2 to 4 in one table. Please improve the presentation and use landscape pages or colour blocks to distinguish different categories if required.
- Please use tables for tables (example: figures 1 to 7 are tables).
- Typo: Naïve Baines Classifier
- Why were the classifiers chosen? What's the sampling strategy? Why? Can other methods help like importance weighting? Please elaborate.
- What's the training-test split? Why? What's the overfitting avoidance strategy here?
In conclusion, the paper's presentation is lacking. Please go over the manuscript and make any necessary changes.
Round 2
Reviewer 3 Report
The authors have performed the necessary corrections to my queries. I recommend acceptance pending a few minor edits:
- Please double-check the flow chart. The data seems to be flowing from coach phone (display) to index (process). Should it be reversed?
- The writing style is poor. Please look into academic writing. I suggest consulting a professional or someone who has published quality journal papers.
Good luck!
Author Response
The flow chart has that mistake, it was fixed.
The writing was checked and review with help from different researchers. The article was improved in writing.